# MOLISENS: A MObile LIdar SENsor System to exploit the potential of small industrial lidar devices for geoscientific applications

Thomas Goelles[2,1*], Tobias Hammer[1,2,3*], Stefan Muckenhuber[1,2,*], Birgit Schlager[2,3,*], Jakob Abermann[1], Christian Bauer[1], Víctor J. Expósito Jiménez[2], Wolfgang Schöner[1], Markus Schratter[2], Benjamin Schrei[1], and Kim Senger[4]

[1]University of Graz - Department of Geography and Regional Sciences, Heinrichstraße 36, 8010 Graz, Austria
[2]Virtual Vehicle Research GmbH, Inffeldgasse 21a, 8010 Graz, Austria
[3]Graz University of Technology - Institute of Automation and Control, Inffeldgasse 21b, 8010 Graz, Austria
[4]University Centre in Svalbard, P.O. Box 156, 9171 Longyearbyen, Norway
[*]These authors contributed equally to this work.

**Correspondence:** thomas.goelles@v2c2.at, hammer.tobias@gmx.de, stefan.muckenhuber@uni-graz.at, birgit.schlager@v2c2.at

**Abstract.** We propose a newly developed modular MObile LIdar SENsor System (MOLISENS) to enable new applications for small industrial light detection and ranging (lidar) sensors. The stand-alone, modular setup supports both, monitoring of dynamic processes and mobile mapping applications based on Simultaneous Localization and Mapping (SLAM) algorithms. The main objective of MOLISENS is to exploit newly emerging perception sensor technologies developed for the automotive
industry for geoscientific applications. However, MOLISENS can also be used for other application areas, such as 3D mapping of buildings or vehicle independent data collection for sensor performance assessment and sensor modeling. Compared to Terrestrial Laser Scanners (TLSs), small industrial lidar sensors provide advantages in terms of size (in the order of 10cm), weight (in the order of 1kg or less), price (typically between $5,000$EUR and $10,000$EUR), robustness (typical protection class of IP68), frame rates (typically 10Hz-20Hz), and eye safety class (typically 1). For these reasons, small industrial lidar
systems can provide a very useful complement to currently used TLS systems that have their strengths in range and accuracy performance. The MOLISENS hardware setup consists of a sensor unit, a data logger, and a battery pack to support stand-alone and mobile applications. The sensor unit includes the small industrial lidar Ouster OS1-64 Gen1, a ublox multi-band active Global Navigation Satellite System (GNSS) with the possibility for Real-Time Kinematic (RTK), and a 9-axis Xsens Inertial Measurement Unit (IMU). Special emphasis was put on the robustness of the individual components of MOLISENS to support
operations in rough field and adverse weather conditions. The sensor unit has a standard tripod thread for easy mounting on various platforms. The current setup of MOLISENS has a horizontal field of view of $360°$, a vertical field of view with a $45°$ opening angle, a range of 120m, a spatial resolution of a few cm, and a temporal resolution of 10Hz-20Hz. To evaluate the performance of MOLISENS, we present a comparison between the integrated small industrial lidar Ouster OS1-64 and the state of the art, high accuracy and high precision TLS RIEGL VZ-6000 in a set of controlled experimental setups. We then
apply the small industrial lidar Ouster OS1-64 in several real world settings. The mobile mapping application of MOLISENS

has been tested under various conditions and results are shown from two surveys in the Lurgrotte cave system in Austria and a glacier cave in Longyearbreen on Svalbard.

## 1 Introduction

Developing new, reliable measurement and monitoring techniques requires emerging cutting-edge technology. This paper introduces a stand-alone, modular MObile LIdar SENsor System (MOLISENS) that builds on recent lidar, radar, and camera innovations from the automotive industry. Originally, these sensors were developed for high-resolution environment perception of automated vehicles. MOLISENS currently includes a lidar, a Differential Global Navigation Satellite System (DGNSS), and an Inertial Measurement Unit (IMU) for georeferenced positioning and orientation. The modular design permits also the use radar and camera sensors (including traffic monitoring sensors). Furthermore, the setup works without the necessity of a complete vehicle setup. This shall allow measuring geoscientific processes reliably, at any remote location, with very high spatial and temporal resolution, and at relatively low costs.

Non-industrial Terrestrial Laser Scanner (TLS) units have been used in many geoscientific applications such as spectral and structural geology, seismology, natural hazards, geomorphology, and glaciology as listed in the review Telling et al. (2017). Data products based on TLS are widely used and range from controlled outdoor studies (e.g., Rapstine et al., 2020; Rengers et al., 2021; Prokop et al., 2008) to natural observations (e.g., Rengers et al., 2021) and damage assessments (e.g., Olsen and Kayen). Typically, multi-temporal lidar studies deal with repeated measurements over years (e.g., O'Neal and Pizzuto, 2011; Neugirg et al., 2016) or months (e.g., Rabatel et al., 2008; Rengers and Tucker, 2015) in order to investigate deforming surfaces. Recently, permanently mounted TLS have been used to monitor rockfall at intervals of an hour (Williams et al., 2018, 2019) or to monitor a high mountain environment at daily or hourly intervals (Voordendag et al., 2021). Rengers et al. (2021) recorded a mass flow with speeds greater than $1m\ s^{-1}$ for the first time. They used a modified VZ-400 TLS at a narrow field of view of $44^{\circ} \times 0.13^{\circ}$, which allowed a sampling rate of 60Hz.

Structure from motion (SfM) especially from Unmanned Aerial Vehicle (UAV), is another tool to derive 3-dimensional (3D) point clouds for geoscientific applications. For example, Lague (2020) compared SfM to TLS for fluvial geomorphology and Wilkinson et al. (2016) for digital outcrop acquisition. Its main advantages are low-cost, uniform point density, high resolution RGB information with the limitations of no penetration through vegetation, requirement of good Global Navigation Satellite System (GNSS) signal, and UAV flight regulations. In addition, SfM is problematic for surfaces with homogenous textures like snow and is limited to well-lit environments. In caves SfM has been used with digital close-range photogrammetry by a digital single-lens reflex camera mounted on a tripod in the study of Pukanská et al. (2020), where they also compared it to TLS data. Their conclusion was that both methods have their specific requirements, advantages, and disadvantages. Therefore, they recommended a combination of both methods for mapping complex cave spaces.

The goal of MOLISENS is to provide an additional tool to SfM and TLS with a unique set of requirements and advantages. It should be cheaper than TLS (currently around $100,000$EUR to $200,000$EUR), with a high sampling frequency and be independent of natural or artificial light, while being small and light enough to carry into tight spaces like caves. It needs to be

suitable for static applications on a tripod as well as for mobile mapping. Furthermore, the mobile mapping approach should
reduce the shadow effects from static TLS uses, all while keeping accuracy and precision high.

Today, the automotive industry is a leading technology driver for small industrial lidar, radar, and camera sensors, because the largest challenge for achieving the next level of vehicle automation is to improve the reliability of the vehicle's perception system (Watzenig and Horn, 2017). lidar, radar, and camera play essential roles in the perception system of automated vehicles (Marti et al., 2019). The presented work will focus on lidar sensors. However, since MOLISENS is designed as modular
system, integration of radar and camera is possible with little effort.

Small industrial lidar sensors record high-resolution point clouds with high acquisition frequencies (around 10Hz-20Hz frame rate) to support applications in fast moving environment such as freeways. High costs of mechanically spinning lidars (currently around $5,000$EUR to $10,000$EUR) are still a limiting factor for many applications, but prices for small industrial lidar have already dropped significantly during the last decade and are expected to drop by another order of magnitude in
the upcoming years caused by newly emerging technologies like micro-electro-mechanical systems (MEMS) based mirrors, optical phased array, single-photon avalanche diode (SPAD) detectors, and vertical-cavity surface-emitting laser (VCSEL) sources (Hecht, 2018; Druml et al., 2018; Thakur, 2016). Examples for state-of-the-art small industrial lidar types are Ouster OS0 (Ouster Inc., 2021a), Ouster OS1 (Ouster Inc., 2021b), Ouster OS2 (Ouster Inc., 2021c), Velodyne Alpha Prime (Velodyne Lidar Inc., 2021), and Ibeo Lux 4L/8L/HD (Ibeo Automotive Systems GmbH, 2021). In addition to range information, several
new lidar types, e.g., Ouster OS (Ouster Inc., 2021a, b, c), provide intensity information for each received point, which allows to take also the reflectance of the illuminated materials for new applications into account. Most state-of-the-art small industrial lidars provide a single return or dual returns, while some have up to five returns (Ocular Robotics Limited, 2018) and one which offers full-waveform information (LeddarTech Inc., 2021). Therefore, this limits the application of small industrial lidar where multiple returns are needed, as for example for vegetation removal.

Due to the above listed advantages, small industrial lidar sensors can complement, non-industrial TLS systems that are nowadays used for geoscientific applications. Examples for state-of-the-art TLS used in geosciences are Leica P30/P40 (Leica Geosystems AG, 2021a), Leica P50 (Leica Geosystems AG, 2021b), or the Riegl VZ–Series (Riegl Laser Measurement Systems GmbH, 2020). High-end TLS typically provide very detailed and highly accurate point clouds but are more expensive (order of $100,000$EUR), heavier (order of 5kg to 10kg), less robust (typically IP64), and in certain field scenarios more difficult
to handle than small and light-weight industrial lidar sensors.

To support mobile mapping applications, MOLISENS includes a DGNSS and an IMU for georeferenced positioning and orientation. For registration of subsequently recorded point clouds into a cumulative point cloud, i.e., for creating a 3D map, the Simultaneous Localization and Mapping (SLAM) algorithm (Bălașa et al., 2021; Zhang and Singh, 2017) LIO-SAM (Shan et al., 2020) is used. Lidar based mobile mapping systems have already been tested in various disciplines such as indoor
mapping applications (Tucci et al., 2018), urban mapping applications (Moosmann and Stiller, 2011; Zhang and Singh, 2017; Behley and Stachniss, 2018), and for geoscientific surveys (Bosse et al., 2012; Kukko et al., 2012; Wang et al., 2013). A major advantage of MOLISENS compared to previous systems is the modular setup focused on small industrial sensors, that allows

to easily exchange and update existing components and extend the system with additional small industrial sensors, such as radar and camera.

Apart from newly emerging perception sensor technology, MOLISENS also benefit from recent developments in the GNSS sector. Typically, DGNSS technology for positioning requires extensive additional gear that must be transported into the field, i.e., rover and base station. New GNSS platforms integrate multi-band GNSS and Real-Time Kinematic (RTK) technology to yield accuracies in the order of centimeters with a single device and internet connection. Such GNSS platforms are designed primarily for industrial tracking and wearable applications so they are optimized in size, weight, update rate, and power 95    consumption (Janos and Przemysław, 2021).

To estimate the quality of small industrial lidar point clouds, a RIEGL VZ-6000 3D high accuracy and precision TLS (Riegl Laser Measurement Systems GmbH, 2020) was used for ground-truth acquisition. A test setup was designed to compare the accuracy of the VZ-6000 to small industrial lidar sensors (Hammer, 2021). In addition, the MOLISENS system has been tested under various conditions and results are shown from two mapping surveys in the Lurgrotte cave system in Austria and a glacier 100    cave in Longyearbreen on Svalbard.

Other potential use cases in physical geography are

– underground measurements: cave/mine mapping,

– monitoring glacier caves, calving glaciers,

– monitoring of snow and avalanches,

– monitoring of mass movements,

– monitoring of fluvial systems,

– monitoring of erosion and deposition processes,

– sea ice detection and mapping,

– coastal mapping,

– archaeology, historical and cultural preservation, and

– forestry surveys.

Potential use cases in urban environments are

– highways surveys,

– roadside inventory projects,

– power line corridor surveys,

- 3D City modeling, and

- 3D indoor modeling.

This article is organized as follows: Section 2 gives an overview on the hard- and software components of MOLISENS. In Section 3, the integrated small industrial lidar OS1-64 is described and compared to the state-of-the-art TLS VZ-6000. In Section 4, the point cloud processing package *pointcloudset* and the used mapping algorithm are described. The results of two mapping campaigns are shown in Section 5. The discussion of the measurement campaigns and an outlook on future applications are given in Section 6. The conclusion is presented in Section 7.

## 2 MOLISENS setup

MOLISENS provides a stand-alone, modular framework which is capable of integrating various small industrial lidar, radar, and camera sensors, that support Robot Operating System (ROS) functionality, with low adjustment effort. The hardware setup follows International Protection (IP) standards of small industrial sensors, e.g., the OS1-64 has an IP class of 69K with the cable attached (Ouster Inc., 2020b), which makes it suitable for fieldwork in rough environments. Figure 1 depicts the hardware components of MOLISENS. The data logger and the sensor unit are connected via a self-developed wire harness which avoids the need of multiple cables. The setup can either be powered by batteries or by a Alternating Current (AC)/Direct Current (DC) mains adapter. The environment is scanned by the sensor unit and the transmitted sensor data are recorded by the data logger. The data can be downloaded via a Local Area Network (LAN) interface for further post-processing on a computer.

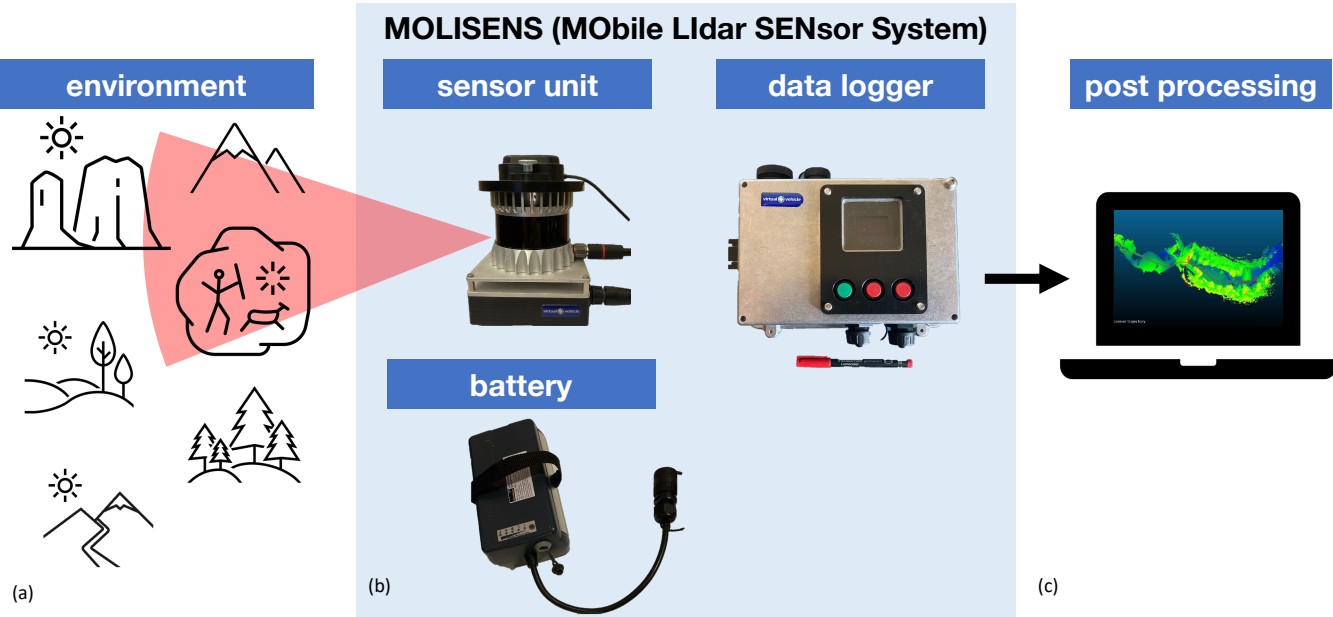

**Figure 1.** MOLISENS concept with hardware components and interfaces (b) with a pen for scale. The data logger is $21.5 \times 16 \times 14$cm and weighs 2.1kg. The sensor unit weighs 1.0kg and the total weight of the complete system is 5.5kg

## 2.1 Sensor unit

The sensor unit consists of the OS1-64 Gen1 which is an small industrial rotating lidar sensor, a ublox active multi-band GNSS antenna of the ANN-MB series, and the 9-axis Xsens MTi 630 IMU. Between the OS1-64 and the IMU is a space that heat produced by the OS1-64 can be dissipated. The sensor unit has a $1/4$ inch thread, which is a standard camera thread, mountable on handles, tripods, or other standardized setups.

## 2.2 Data logger

### 2.2.1 Hardware

The data logger consists of two DC/DC converters, one $24$V/$24$V converter and one $24$V/$5$V converter for internal power supply, a RaspberryPi 4 as processing unit, a RaspberryPi Hardware Attached on Top (HAT) for the real-time clock, a RaspberryPi HAT with a 1TB Solid State Drive (SSD) for data storage, a RaspberryPi HAT for GNSS data, a Long Term Evolution (LTE) stick to retrieve RTK data and the Ouster interface board that is responsible for powering the OS1-64 and for data transmission. Interfaces provided by the data logger are a connector for power supply of the whole setup, a 24-pin connector for Ouster data and power supply, the IMU Universal Serial Bus (USB) interface, a Sub-Miniature A (SMA) connector for the GNSS antenna, a Registered Jack 45 (RJ45) connector for Ethernet, and a USB connector. Furthermore, the data logger's interface includes one on/off button, two red buttons for selecting the measurement programme, one green button for start and stop measurements,

and an Organic Light Emitting Diode (OLED) display, which shows the measurement programs, the state of the LTE connection, and the filename of the current measurement. The aluminum housing of the data logger includes an aluminum plate to the integrated circuits on the RaspberryPi which allows appropriate cooling of the hardware. The OLED display can be seen through a transparent plastic window in the aluminum housing.

### 2.2.2 Software

The software stack of the data logger is shown in Figure 2. Although the official operating system for RaspberryPi is Raspbian (https://www.raspbian.org/), Ubuntu Server 20.04 Long Term Support (LTS) (https://ubuntu.com/download/raspberry-pi) was installed as operating system for MOLISENS, since the integration with ROS (https://www.ros.org) works best on Ubuntu. ROS is an open-source middleware widely used for robotic applications. A major advantage of ROS is the extensive list of open-source third-party packages and tools for different domains, e.g., Autoware (https://www.autoware.org/) for the automotive domain.

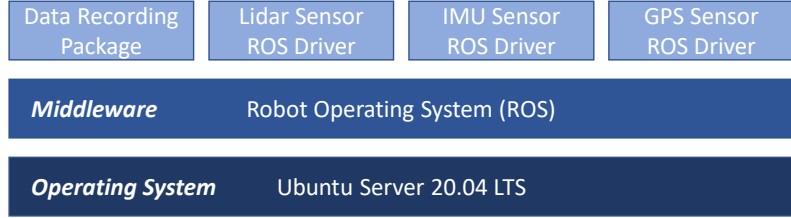

**Figure 2.** Software stack of the data logger.

In ROS, a master called roscore controls and registers all nodes running in the system. Each node, which can be defined as an entity that performs a task, can exchange data with other nodes by publishing or subscribing messages through topics. Topics are communication channels, which are defined by a unique name and a specified type of message that is transported. ROS officially supports C++, Python, and Lisp but other programming languages are also possible through unofficial channels.

The specific software packages used in MOLISENS are:

– Data recording package: We developed a ROS package in Python that provides an easy interface to start and stop the data recording as well as a flexible configuration for the specific requirements of the use case. Both, the sampling rate of either 10Hz or 20Hz of the OS1-64 and the number of points in horizontal direction of either 1,024 or 2,048, can be selected by the user just before the measurements by using the red and green buttons on the data logger. The IMU data are recorded with 200Hz and the GNSS data with 1Hz. The ROS package records all messages from the specified topics and creates a time synchronous ROS bag file including the data from all sensors. Later, this ROS bag file can be used as input to a SLAM algorithm to generate a 3D map of the measurement area.

- Lidar sensor package: The ROS package provided by the sensor manufacturing company was implemented (Ouster Inc., 2020a). It provides transforming the raw data from the sensor into point cloud messages and also includes visualization tools to proof that the lidar sensor is correctly mapping the scenario and to check the light intensity of the points. Due to the computational limitations of the RaspberryPi, only raw data are recorded. The recorded raw data are converted into point cloud messages in a post-processing step.

- IMU sensor package: Similar to the lidar sensor, the ROS driver provided by the manufacturer is used (Xsens, 2021). Only the configuration and topic selection was adopted to meet the requirements of our use case. Most of the topics were omitted to increase the performance of the system.

- GNSS sensor package: Another self-developed Python package was used to retrieve the National Marine Electronics Association (NMEA) messages from the ublox GNSS module. This driver is also able to receive correction data from Radio Technical Commission for Maritime (RTCM) messages through the integrated Networked Transport of RTCM via Internet Protocol (NTRIP) client. By using this NTRIP client, correction data from the external services are included into the GNSS module to improve the accuracy of the measurements. The usage of this correction data is called GNSS RTK. The precision is below 2.5cm with fixed RTK when the signal from the satellites is clear and also the base station from the correction data service is not far away, i.e., less than 10km, from the GNSS module (Dunning, 2018). When the situation is not optimal, the module is still able to reach a precision between 10cm and 45cm with RTK float (Dunning, 2018).

## 2.3 Power supply

MOLISENS can be powered either with batteries (e.g., Lithium Iron Phosphate (LiFePO4), Lithium-Ion (Li-ion)) or with an AC/DC mains adapter that provides a nominal voltage of 24V. The battery supply supports mobile measurements whereas the mains adapter may be used when recorded data are transferred to the post-processing computer. The described setup including the data logger and the sensor unit draws a current of about 1A when data of all three sensors are recorded. We used either a Li-ion-battery with 10.4Ah for 10.4h of measuring or two parallel LiFePO4-batteries with 3.6Ah each, so 7.2Ah in total, for 7.2h of measuring. The operating temperature for discharging is limited between $-20°C$ and $+60°C$ for both types of batteries. We used batteries from AccuPower (AccuPower Research, Development and Distribution Company (Ltd.), 2022).

## 3  MOLISENS with small industrial lidar OS1-64 Gen1

### 3.1  OS1-64 Gen1 specifications

The Ouster OS1-64 is a mechanical spinning lidar scanner that costs about 10,000EUR. The ingress protection level is IP69K with Input/Output (I/O) cable attached, so it offers complete protection against contact, i.e., it is dust-tight and waterproof. It is classified as a mid-range lidar sensor and can detect objects up to a distance of 150m. The minimum range is 0.8m. The laser operates with eye safety class 1 per IEC 60825-1:2014, which makes it possible to operate the lidar without any restrictions

regarding the eye safety of the operator or other persons within the measurement range. The wavelength of the laser is 855nm. The range resolution is 0.3cm, so it is able detect individual objects when the distance between those objects in scanning direction is 0.3cm or greater. The range accuracy is stated with +/-5cm for Lambertian targets and +/-10cm for retroreflectors (Ouster Inc., 2020b). The precision depends on the range and is between +-1cm and +-5cm. The vertical resolution of the OS1-64 is given by the 64 channels which are evenly distributed within the 33.2° vertical field of view. The horizontal resolution is configurable and can be 512, 1,024, or 2,048 scanning points in horizontal direction for the 360° field of view. The angular sampling accuracy vertically and horizontally is +/-0.01°. The sampling frequency can be configured with 10Hz or 20Hz. At 10Hz, the scanner rotates and scans 10 times per second and produces up to $64 \cdot 2,048$ points per rotation. Therefore, the OS1-64 is able to detect over 1.3 million points per second at 10Hz rotation rate.

A computer with ROS installed can read and record the data which are forwarded by the Ouster interface box. In our case, the necessary components are integrated in MOLISENS. We decided to record only raw lidar data to be able to store more data at about 1.3 million points per second, IMU measurements with 200Hz, and GNSS measurements with 1Hz. The raw lidar data do not comprise the actual 3D-points with x-, y-, and z-coordinates, but information such as timestamp, measurement id, and range for each measurement. These data is used in a post-processing step for the derivation of 3D-points. Recording the raw data instead of the point cloud data reduces the total data rate, i.e., lidar, IMU, and GNSS data, from 77MB/s to 15.14MB/s, a factor of more than 5. Up to 18h of recording are possible with 1TB of data storage that we use in MOLISENS.

### 3.2 OS1-64 performance assessment

We assessed the performance of the OS1-64 against the VZ-6000 in a standardized test setup which is based on Boehler et al. (2003). The first frame of each OS1-64 measurement was used for the test. Specifically, the following attributes were analyzed: systematic and surface-induced range errors and angular errors.

We tested the systematic noise in the range measurements of the lidar. For that purpose, a plane target was scanned. We used a wall perpendicular to the observation direction as the plane target. The wall was modeled as a plane and the normal distance between every point in the point cloud and this plane can be calculated. The standard deviation of the distribution of these normal distances, which represent the range errors, was derived (Figure 3). We investigated two different materials: retroreflective foil and a cardboard with black dull spray paint. Those surfaces need to be in the same plane perpendicular to the observation direction to quantify the deviations of surfaces with high and low reflectivity. This was realized by attaching the materials to a wooden board which was mounted on the wall. We analyzed the reflectance of the materials based on the comparison to Lambertian targets. From a perpendicular angle the reflectivity of the retroreflector was 200% relative to a 100% Lambertian target. The black dull spray paint had a reflectivity of 10% relative to a 100% Lambertian target (Muckenhuber et al., 2020, Birkebak et al., 2018). The scanned wall in the background was used to model a reference plane. The normal distances between points and reference plane can be calculated by subtracting the thickness of the wooden board and the target thickness. A threshold in the reflectance value was used to select the points representing the target in the pointcloud.

We used scanned circles to determine the vertical and horizontal distances for quantifying the angular accuracy. We attached four circles, representing a rectangle, with dimensions of $4.5\text{m} \cdot 2\text{m}$, to a wall. This rectangle was scanned from three different positions which yields six independent vertical and six independent horizontal distances.

The results of the tests on systematic range errors showed that the OS1-64 has a higher standard deviation in the range error distribution compared to a TLS such as the VZ-6000 (Figure 3 (a)). The most significant range errors of up to 25cm occurred when scanning retroreflective targets with the OS1-64 (Figure 3 (d)). Furthermore, the range errors in this case are not only larger but are also spread out over a large range of values with a standard deviation of 6.9cm. According to the manufacturer, the errors occurring with retroreflectors result from the time walk error. This error is caused by clock errors and is an internal error source of lidar systems. This means that the light returns so strongly that it deforms the shape of the received signal which leads to an error in the estimation of the peak, i.e., the distance measured (Nahler et al., 2020).

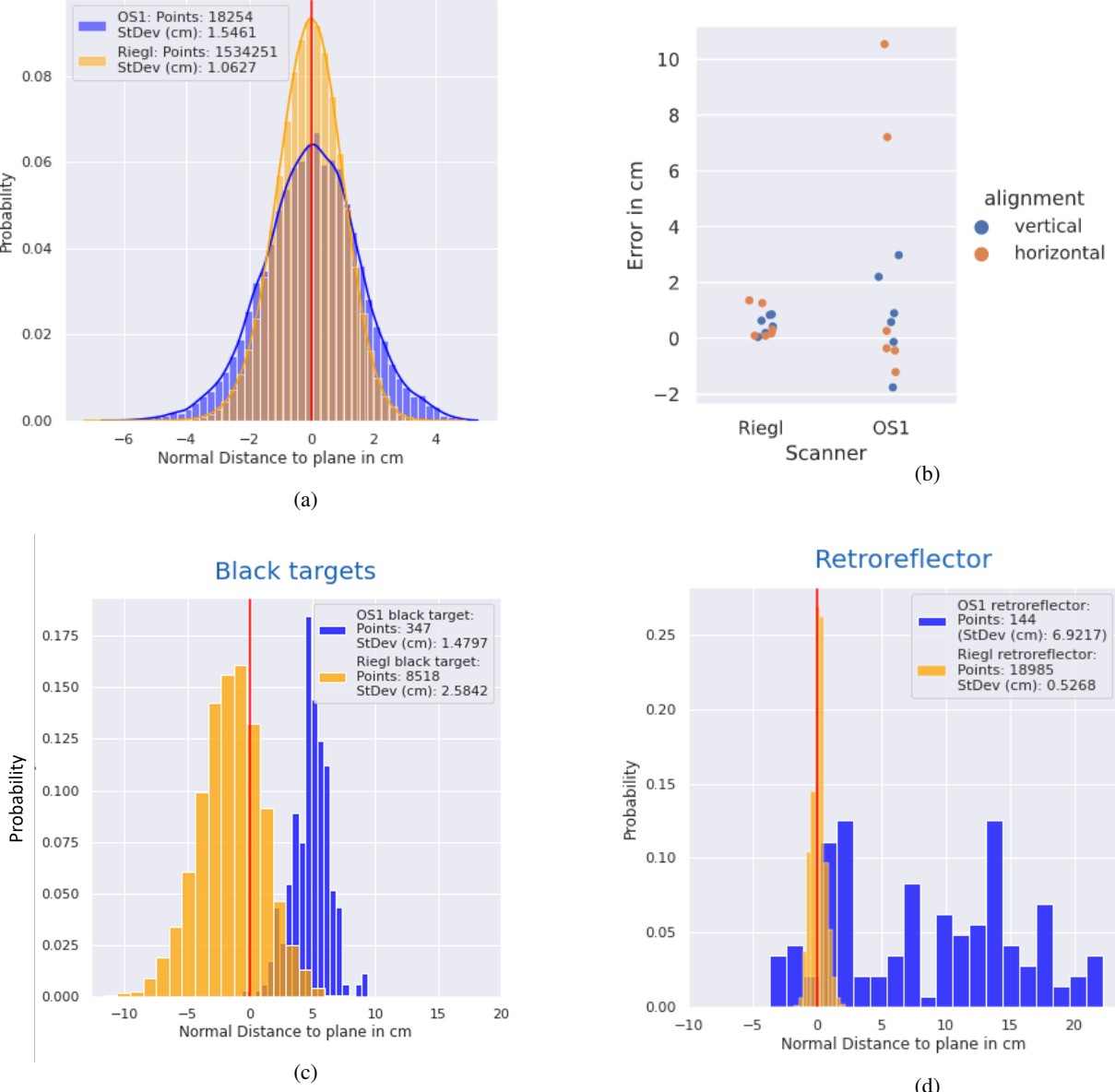

**Figure 3.** Results of the accuracy assessment of the OS1-64 compared to the VZ-6000 which was used as the reference lidar: (a) range error for a white wall, i.e., normal distances between measurements and ideal plane of the wall, (b) angular error, (c) range errors for black target with low reflectivity, i.e., normal distances between measurements and ideal plane of black target, (d) range errors for retroreflectors, i.e., normal distances between measurements and ideal plane of retroreflector.

Figure 4 shows parts of point clouds with observed artifacts. Areas without data appear near highly reflecting objects (Figure 4 (a) and (b)). The results also show that the OS1-64 performed better at detecting low reflectance targets in close proximity than the VZ-6000 (Figure 4 (c) and (d)).

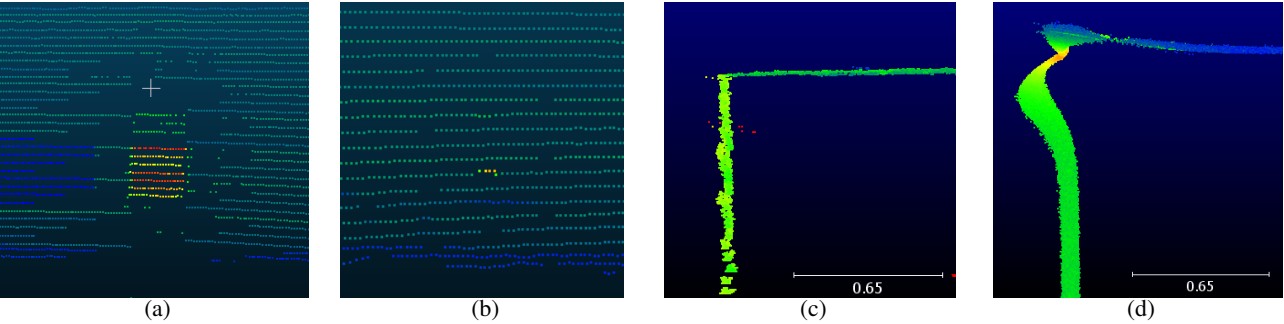

**Figure 4.** Artifacts observed in the accuracy tests: (a) and (b) areas without data resulting from blooming filter in the OS1-64 data, (c) a part of the OS1-64 point cloud with a corner of the room, (d) a part of the VZ-6000 point cloud with a corner of the room.

The found performance issues comprise the following artifacts: range errors at highly reflective targets for the OS1-64, areas without data around highly reflective targets for the OS1-64, corner artifacts for the VZ-6000, and multipath artifacts for the VZ-6000. The systematic range errors, which can be considered as noise, can have great impact on mapping micro-scale features. According to the manufacturer, the effect of areas without data is caused by an integrated blooming filter in the OS1-64. This filter removes bloomed, i.e., overly saturated, points. The range errors of the OS1-64 have a big impact on current point cloud georeferencing methods where retroreflectors of known position are used as distinguishable objects in a point cloud. Other georeferencing methods, like the matching with georeferenced point clouds, must be considered.

## 4 Data processing for lidar applications

### 4.1 Data processing of point cloud time series

Small industrial lidar sensors record point clouds in the order of a million points per second at acquisition rates ranging from 10Hz to 20Hz. In addition to the $x$-, $y$-, and $z$- coordinates, also range, intensity, reflectivity, ambient near infrared, azimuth angle, and time stamp for each point are stored (Ouster Inc., 2020b). This leads to datasets in the order of gigabytes in the form of 3D point clouds recorded over time. Most software deals with single point clouds and the tools provided by ROS are not designed for post-processing and data analytics. Therefore, a Python package called *pointcloudset* (Goelles et al., 2021) was developed along the sensor hardware. The package is available on the Python Package Index (PyPI) for easy installation with pip. This package organizes the data in the following way: The point cloud data stored in a ROS bag-file (.bag) is read into a *pointcloudset Dataset*. This *Dataset* object consists of multiple *PointCloud* objects, timestamps, and metadata.

The package is optimized for analytics on the whole dataset to answer questions like: "At which point and when was the highest returned intensity of the whole dataset?" or "How many clusters of points in a 0.5m radius exist between 5 and 10m in x direction in the 124s frame?". Several queries like this can be chained together to form complex pipelines. The computation is only executed at the very last step when the answer is required by so called "lazy evaluation". The computation is performed on multiple Central Processing Units (CPUs) in parallel. The package is not limited to built-in functions and additional arbitrary

functions can be implemented and applied. Furthermore, the package provides tools for visualization, import and export of widely used point cloud formats. Also a direct interface to the powerful open3D and pandas libraries (Zhou et al., 2018; The pandas development team, 2020) is implemented for additional applications. For more details see the documentation on https://virtual-vehicle.github.io/pointcloudset/.

## 4.2 SLAM algorithm

In robotics, SLAM algorithms are a fundamental prerequisite for feedback control, obstacle avoidance, and planning since SLAM allows a robot's six Degrees Of Freedom (DOF) state estimation (Bălașa et al., 2021). Here, we use a SLAM algorithm to generate one cumulative point cloud from a time-series of point clouds. MOLISENS is either mounted on a moving platform or carried along by a person while recording data. The data recording unit uses ROS as middleware and all data is recorded in a ROS bag file, which includes IMU, lidar, and GNSS data. Each recorded data type in the ROS bag file has a timestamp. The recorded data is the input for the mapping algorithm LIO-SAM (Shan et al., 2020), which is applied offline in a post-processing step.

LIO-SAM uses the lidar odometry data to estimate the six DOF trajectory of the mapping sensor. The state estimation problem is solved by a factor graph. This incorporates IMU-pre-integration, lidar odometry, GNSS data, and loop closure. The system does not depend on continuous GNSS data. Therefore, the GNSS factor is only added when the estimated position covariance is larger than the received GNSS position covariance. The loop closure factor is responsible for detecting whether a new node has a small Euclidean distance to a prior state. If this is detected, the algorithm tries to match the new state to the near, past state. This is especially useful to correct for potential drift in altitude when GNSS is the only absolute sensor available. These advantages compared to methods such as Lidar Odometry and Mapping (LOAM) (Zhang and Singh, 2017) and other previous state of the art algorithms made it well suited for our use cases.

## 5 Applications in geoscience

To test the MOLISENS setup in challenging field conditions, two mapping surveys in the Lurgrotte cave system in Austria and in a glacier cave in Longyearbreen on Svalbard have been conducted. The following Section presents the results of these two measurement campaigns.

### 5.1 Application in speleology

The Lurgrotte, a partially water-bearing cave 15km north of Graz in Styria, Austria, was chosen as a study area for MOLISENS. The approximately 6km long cave passes through the Tanneben massif between the localities of Semriach and Peggau. Parts of the cave are accessible to tourists. The cave is characterized by an abundance of speleothems, water-bearing passages, and a heterogeneous cave geometry in which narrow passages alternate with large chambers, such as the Great Dome. With an area of approximately $5,100\text{m}^2$, the Great Dome is one of the ten largest cave chambers in Austria (Plan and Oberender, 2016).

Due to these heterogeneous characteristics, the Lurgrotte Semriach is well suited for testing the application of MOLISENS in speleology.

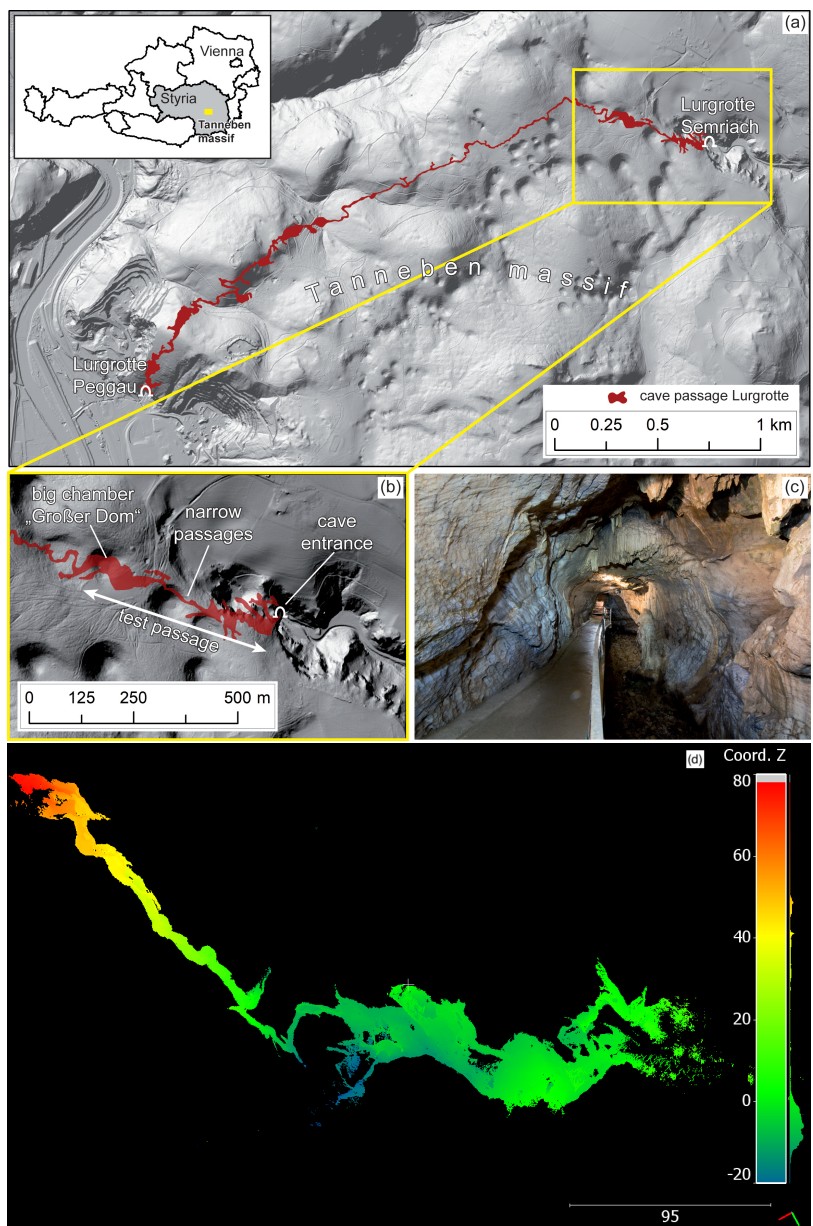

**Figure 5.** Study area Lurgrotte Semriach. (a) Hillshade visualization (1m · 1m raster resolution) and cave map (b) detailed view of the test passage in which the OS1-64 surveys were carried out including the Great Dome (Großer Dom) (c) a narrow passage with paved footpath within the cave segment open for visitors (image by Christian Bauer). Data: cave map: Bock and Dolischka (1953); ALS-data: CC-BY-4.0: Land Steiermark - data.steiermark.gv.at. (d) Measurement (2), starting at cave entrance, turning point right before the Great Dome, and end at cave entrance; point cloud colored by z-coordinate in metres, visualized with CloudCompare.

Scanning a cave system, such as the Lurgrotte Semriach, with a TLS would demand tens to hundreds scan positions, i.e., a significant effort in terms of time and costs. Using MOLISENS, we were able to produce a point cloud without the necessity of time consuming scans at individual positions. The mapping campaign demonstrated that MOLISENS can provide a cumulative point cloud even without the use of GNSS measurements. Also, the LIO-SAM algorithm was tested on whether it is able to co-register point clouds that were recorded partly outdoor and indoor. More than 300m of complex cave geometry could be scanned with MOLISENS in less than 12 minutes (Figure 5 (d)). Measurement (1) includes the switch from an outdoor environment to an indoor environment in a single measurement. Measurement (2) was conducted only inside the cave. An overview of the recorded data is given in Table 1.

**Table 1.** Comparison of measurements conducted in Lurgrotte; 0.1m is the minimum average point spacing that can be produced by LIO-SAM

| Parameter | Measurement (1) | Measurement (2) |
|---|---|---|
| Duration | 6min 37s | 11min 58s |
| IMU messages | 158,816 | 287,525 |
| Lidar packet messages | 499,368 | 912,505 |
| Size of .bag file | 6.0GB | 10.9GB |
| Number of points in final point cloud | 10,719,001 | 6,393,490 |
| Average point spacing | 0.1m | 0.1m |

It has to be noted that a final validation of the point cloud data was not possible at this stage. A validation of the data quality and accuracy requires a geodetic reference. A marked closed traverse and local reference points were measured from the cave entrance to the Great Dome after our fieldwork at the Lurgrotte. The drilled mountings of these marks could be used used again for further tests with our system. A valid assessment of the accuracy of the produced map can then be accomplished with this reference.

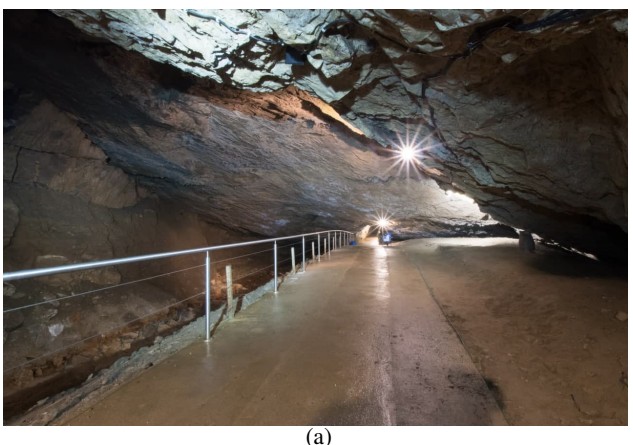
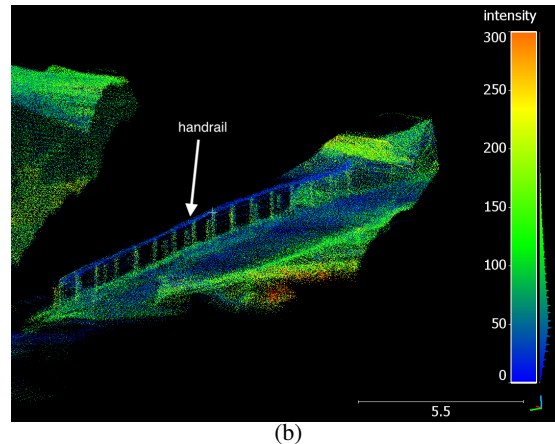

(a)                                              (b)

**Figure 6.** Panel (a) shows karst shapes, concrete, and metal structures in the Lurgrotte cave (image by Christian Bauer). Panel (b) shows a detailed image of the cave section including the walking path and handrail (created with CloudCompare).

## 5.2 Application in glaciology

To test MOLISENS for cryospheric applications, a glacier cave was mapped in the glacier Longyearbreen on the Svalbard
315  archipelago, Norway. The morphological changes of glacier caves give information about the englacial water routing. Ice volume changes in caves are common throughout the year and the inter-seasonal comparison of ice dynamics can indicate a change in the hydro-climatic regime of the glacier (Perşoiu and Lauritzen, 2017). Previous work on cold glacier caves in the study area involved geomorphological mapping and seasonal temperature monitoring (Alexander et al., 2020; Guðmundsdóttir, 2011), but detailed 3D measurements of a glacier cave system are typically not available.

320  Our aim was to create a 3D point cloud which represent the shape of the glacier cave and the surrounding surface of the glacier. The results are shown in Figure 7. The measurement campaign showed that it is possible to create a cumulative point cloud from the predominant surfaces in and around glacier caves. These surfaces are composed of ice, snow, sediments, and moraine material. Measurements were recorded by walking through the caves bidirectionally with MOLISENS. The recorded data resulting were then processed with the LIO-SAM algorithm to create a cumulative point cloud of the glacier cave and the
325  surrounding surface of the glacier.

Figure 7 (c) represents a segment of the processed point cloud showing the glacier cave and the glacier surface from below. The cross sections shown in Figure 7 (d) are one meter long segments along the cave direction. Some outlier points are visible that might be a result of the scanners range errors and the torso of the person holding the scanner. In general, the Ouster OS1-64 should perform well for glaciological applications due to the wavelength of 855nm where absorption in ice is lower (Warren
330  and Brandt, 2008) than at the wavelength of the VZ-6000 at 1064nm or other TLS which have a wavelength of typically 1550nm (Deems et al., 2013). Further investigations of the errors are needed in the future.

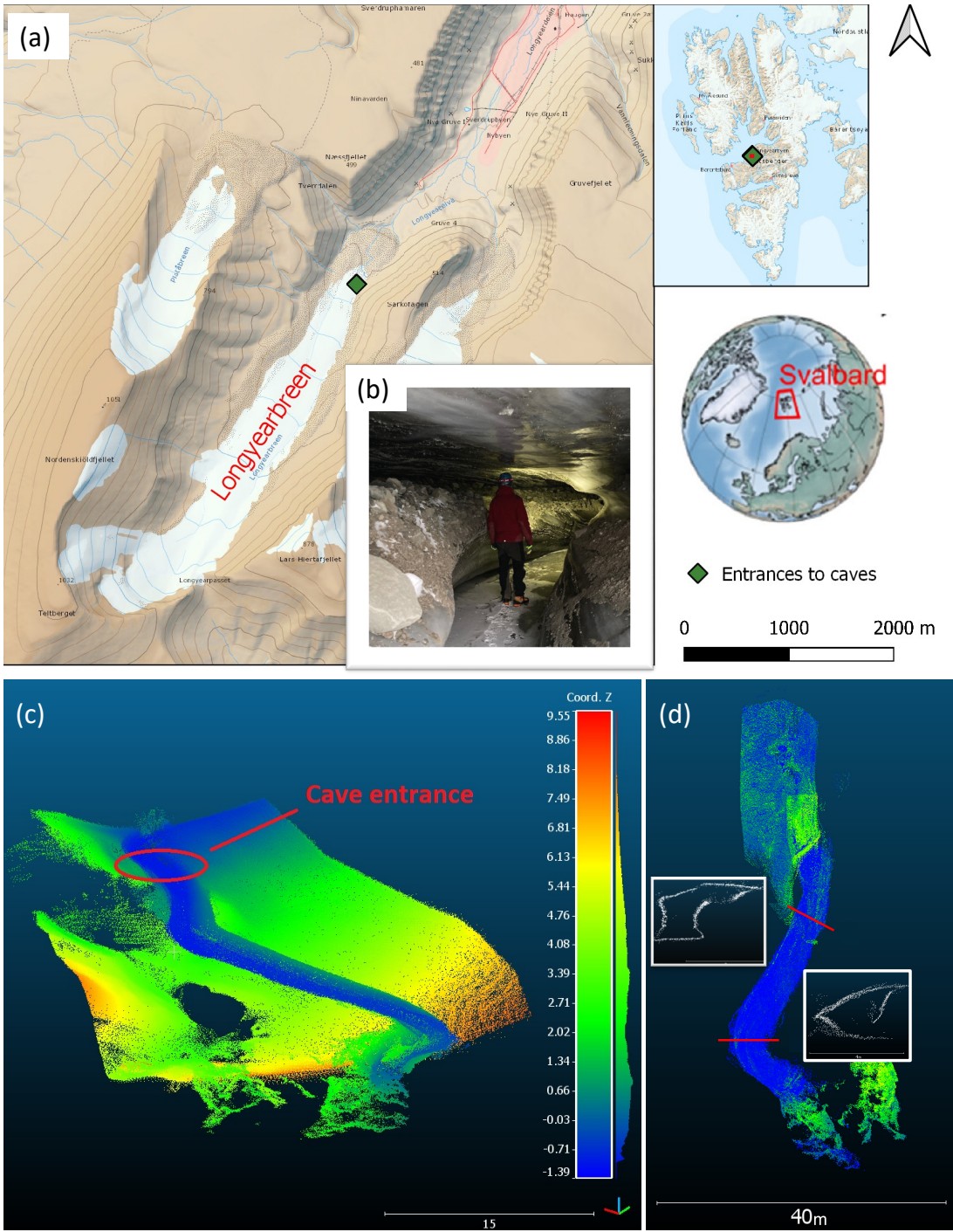

**Figure 7.** Resulting point cloud visualizations of the glacier cave on Longyearbreen (a). Panel (b) shows and image of the entrance and (c) a view from below the point cloud with the glacier cave and the glacier surface. Panel (d) shows the extracted cave (nadir view) with two cross sections with the same color bar as in (c).

## 6 Discussion and outlook

### 6.1 Small industrial lidar as part of MOLISENS

Our comparison shows that small industrial lidar sensors such as the OS1-64 can offer advantages for certain applications compared to conventional TLS such as the VZ-6000. These advantages are the lower price, smaller size, lower weight, and increased robustness but also the ability to acquire data in narrow spaces and while moving. We found that the VZ-6000 shows larger errors when scans are conducted in indoor environments and at low ranges between 5 and 15m.

The accuracy of the data depends on the target properties. A black target had errors of up to 5cm, while a retroreflective material up to 22cm. Nevertheless, for numerous applications the data can be the foundation for many kinds of analysis. We recommend to refrain from analyzing features smaller than 10cm in the processed point clouds. The given accuracy of up to 5cm for ranges greater than 50m leads to an increase in noise when the SLAM algorithm is used. Applying smoothing filters on the cumulative point cloud is recommended. The frequency and magnitude of the distance error when scanning retroreflective surfaces is expected to be reduced with upcoming firmware version. Although, this must be considered if retroreflective targets are used during geo-referencing. Better yet would be to use non-retroreflective target for geo-referencing, e.g., white paper with black markings. Another challenging surface is water, which absorbs typical wavelengths of small industrial lidar in the near-infrared (e.g., Lague, 2020).

In our fieldwork, MOLISENS has proven to record data in complex environments even without a GNSS signal. Although, the drifts in the SLAM processed point clouds has yet be quantified and workflows for georeferencing have to be tested. With the data from our fieldwork, the mapping algorithm LIO-SAM was able to map the environment and the trajectory of the mapping sensor into a point cloud. This shows that results are possible under the following conditions:

– snow and ice surfaces,

– arctic weather conditions ($-20°$C),

– very narrow spaces ($< 1$m), and/or

– rough sensor handling due to rough terrain or narrow spaces.

The lack of GNSS data for cave measurements caused drifts induced by small propagating errors in the IMU data. These drifts are yet to be quantified.

### 6.2 Other small industrial sensors

In addition to lidar, other small industrial perception sensors such as radar systems can be integrated into MOLISENS. Modern automotive and traffic monitoring radar sensors typically operate at 24GHz, e.g., Smartmicro TRUGRD Stream (s.m.s, smart microwave sensors GmbH, 2021), or 77GHz (Ramasubramanian and Ramaiah, 2018), e.g., Continental ARS540 (Continental AG, 2016, 2017), have a range up to 300m, and apply Frequency-Modulated Continuous Wave (FMCW) technologies for

relative distance and velocity estimation (Patole et al., 2017) and digital beam-forming to control the direction of the emitted wave (Hasch, 2015). In addition to data on object level (i.e., list of detected traffic participants), radar data is typically also provided as radar clusters. Clusters represent radar detections with information like position, velocity, and signal strength. This raw data format allows to develop and apply new algorithms for detecting changes in the backscatter behavior of the environment caused by various geoscientific processes.

## 6.3 Potential applications

We envision MOLISENS as a useful tool in geosciences. The IP level of the OS1-64 allows us to conduct measurements under adverse conditions and rough sensor handling. This opens a wide range of applications ranging from cave mapping, glacier surface analysis to meltwater channel monitoring with the potential to increase our understanding of the drainage systems of glaciers. A 3D model of a glacier cave can be used to parameterize volumetric properties of the cave with the aim of analyzing cave morphology (Gallay et al., 2015; Šupinský et al., 2019). In addition, different surface types can be distinguished if they show a significant difference in intensity. E.g., the recorded intensity values for ice surfaces are significantly lower than for surfaces covered with, e.g., moraine material or sediments. This allows to distinguish bare ice from gravel based surface covers.

The portable nature, low cost, and robustness of MOLISENS opens up for new applications well beyond cave mapping. Mobile high resolution 3D mapping of glacier fronts using snowmobiles on sea ice is another possible application and could be conducted at relatively high velocities (up to 60-80km/h). Similarly, regular mapping of coastal bluffs susceptible to coastal erosion (e.g., Guégan and Christiansen, 2017), can be undertaken throughout the polar night season that hinder SfM photogrammetry for large parts of the year in polar regions.

Besides mobile measurements, static measurements can be conducted with MOLISENS to record rapid processes in 3D over time with up to 20Hz. The scanner can be placed permanently in an area of interest and in case of an event, the scanning process could be initiated automatically or remotely. Vice versa constant scanning could detect a process happening which would trigger further process chains.

MOLISENS is also a handy teaching tool since it rapidly acquires data at a fraction of the cost of a conventional TLS. In addition, it can be taken along on excursions more easily, and safety concerns are minimal even in large groups due to the laser class 1 rating. At the University of Graz and the University Centre in Svalbard (UNIS) it is planned to use MOLISENS for excursions and practicals focusing on cryospheric topics, mapping methods, integrated geological methods, and digital geological techniques (Senger et al., 2021).

This list can be further extended since the system can be attached to a wide range of platforms. Tests have been conducted with platforms like cars, agricultural machines, and boats with promising results. Further optimizing weight and power consumption of the system can also enable small UAVs as potential platforms. Apart from geoscientific applications, MOLISENS provides an easy to use setup for testing automotive perception sensors for, e.g., sensor modeling and sensor Fault Detection, Identification, and Recovery (FDIR) method development.

## 7 Conclusion

In this work, we present a newly developed mobile lidar sensor system called MOLISENS. The system combines an small industrial lidar with IMU and GNSS. It provides the opportunity to collect 3D data for a wide range of use cases and applications. Besides the hardware we introduced the post-processing tools provided by the two open source packages LIO-SAM and *pointcloudset*. LIO-SAM is a SLAM algorithm for cumulative point cloud generation, and *pointcloudset* a Python package for analysis and post-processing of static measurements.

The integration of the small industrial lidar OS1-64 and the mobile mapping approach was tested in measurement campaigns in the Lurgrotte cave, Austria and in glacier caves on Longyearbreen, Svalbard. The system offers a flexible, easy to use, and time-efficient way to acquire 3D point cloud, GNSS, and IMU data. The offline SLAM processing resulted in point clouds which can be the basis to investigate numerous geoscientific problems. The robustness of the sensors and the data logger as well as the battery and storage capacities are well suited to demanding fieldwork situations.

In the near future, additional sensors, such as radar and camera, shall be integrated into MOLISENS and further broaden the range of applications. This is possible due to the modular design structure of MOLISENS.

*Code and data availability.* pointcloudset is available at https://github.com/virtual-vehicle/pointcloudset and LIO-SAM at https://github.com/TixiaoShan/LIO-SAM

*Sample availability.* Point could data from Longyearbreen Glacier Cave, Svalbard is available at https://doi.org/10.3217/182j2-hdn17

*Competing interests.* No competing interests are present.

*Acknowledgements.* The publication was written at the University of Graz and at the Virtual Vehicle Research GmbH. The authors would like to acknowledge the financial support within the COMET K2 Competence Centers for Excellent Technologies from the Austrian Federal Ministry for Climate Action (BMK), the Austrian Federal Ministry for Digital and Economic Affairs (BMDW), the Province of Styria (Dept. 12) and the Styrian Business Promotion Agency (SFG). The Austrian Research Promotion Agency (FFG) has been authorised for the
programme management. A special thanks to Oliver Mariani from Virtual Vehicle Research GmbH for supporting the construction of several hardware parts of MOLISENS. The authors would like to acknowledge Andreas Schinnerl for providing access to the Lurgrotte Semriach and supporting the measurements. The fieldwork in Svalbard was in part funded by the Svalbard Science Forum Arctic Field Grant. The authors acknowledge the financial support by the University of Graz.

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
