# Peer review of "MOLISENS: A MOBILE LIDAR SENSOR SYSTEM TO EXPLOIT THE POTENTIAL OF SMALL INDUSTRIAL LIDAR DEVICES FOR GEOSCIENTIFIC APPLICATIONS"

_Geoscientific Instrumentation, Methods and Data Systems, 2022_

## Author Comment (AC1)

**Referee 1**

General Comments:

This is an interesting manuscript that aims to illustrate how inexpensive lidar units developed for the automotive industry can be used for geoscience applications. The authors do a nice job at outlining the technical aspects of the lidar units they compare.

My general suggestions for improvements to the paper are to do a better job of setting up the problem in the introduction. In particular, can you better describe why users would want to use these systems versus techniques like photogrammetry. Your underground examples make it obvious as to why you need lidar, but I think you need to clearly explain that to readers. In addition, you should expand the intro to show the breadth of how people have been using lidar in the geosciences ranging from controlled outdoor studies (e.g., Rapstine et al. 2020; Rengers et al., 2021) to natural observations (e.g., Rosser et al. 2005) to damage assessments (Olsen and Kayen, 2013).

Another rather large suggestion is to try to gear the paper to seem relevant far into the future. Right now, there are references in euros and references to years when technology is expected to be developed, but those references will seem irrelevant 10 years from now. If you could re-frame the tone to have a long-view (at least a decade) I think it will seem more relevant.

In addition to these general suggestions, I have provided several line comments below.

Olsen, M. J., & Kayen, R. (2013). Post-earthquake and tsunami 3D laser scanning forensic investigations. In *Forensic Engineering 2012: Gateway to a Safer Tomorrow* (pp. 477-486).

Rapstine, T. D., Rengers, F. K., Allstadt, K. E., Iverson, R. M., Smith, J. B., Obryk, M. K., ... & Olsen, M. J. (2020). Reconstructing the velocity and deformation of a rapid landslide using multiview video. Journal of Geophysical Research: Earth Surface, 125(8), e2019JF005348.

Rengers, F. K., Rapstine, T. D., Olsen, M., Allstadt, K. E., Iverson, R. M., Leshchinsky, B., ... & Smith, J. B. (2021). Using High Sample Rate Lidar to Measure Debris-Flow Velocity and Surface Geometry. Environmental & Engineering Geoscience, 27(1), 113-126.

Rosser, N. J., Petley, D. N., Lim, M., Dunning, S. A., & Allison, R. J. (2005). Terrestrial laser scanning for monitoring the process of hard rock coastal cliff erosion. *Quarterly Journal of Engineering Geology and Hydrogeology*, *38*(4), 363-375.

Dear Editor, dear reviewers,

We are very grateful for two very detailed and constructive reviews and appreciate the valuable time put into this. We believe by incorporating the reviews we managed to achieve a much more mature manuscript which we hereby submit for your consideration.

In the following we mark black the comments given by the reviewers and the editor, give our answers and comments in blue and indicate how we addressed the amendments in the manuscript in green.

We wrote a new paragraph introducing TLS applications in geosciences, which includes the listed references and more.

„Non-automotive Terrestrial Laser Scanner (TLS) units have been used in many geoscientific applications such as spectral and structural geology, seismology, natural hazards, geomorphology, and glaciology as listed in the review Telling et al. (2017). Data products based on TLS are widely used and range from controlled outdoor studies (e.g., Rapstine et al., 2020; Rengers et al., 2021; Prokop et al., 2008) to natural observations (e.g., Rengers et al., 2021) and damage assessments (e.g., Olsen and Kayen). Typically, multi-temporal lidar studies deal with repeated measurements over years (e.g., O'Neal and Pizzuto, 2011; Neugirg et al., 2016) or months (e.g., Rabatel et al., 2008; Rengers and Tucker, 2015) in order to investigate deforming surfaces. Recently, permanently mounted TLS have been used to monitor rockfall at intervals of an hour (Williams et al., 2018, 2019) or to monitor a high mountain environment at daily or hourly intervals (Voordendag et al., 2021). Rengers et al. (2021) recorded a mass flow with speeds greater than $1m\,s^{-1}$ for the first time. They used a modified VZ-400 TLS at a narrow field of view of $_{\cdot\cdot}44 \times 0.13$ , which allowed a sampling rate of 60Hz.

Concerning the future proofness of the paper: We deleted speculative releases of lidar devices. Although, we kept current prices of units since we think it is relevant because lidar from the automotive industry are much cheaper than "conventional" TLS units. This enables new applications where the lidar unit is at risk of damage or loss.

We added a paragraph motivating the development of MOLISENS. This and the new paragraphs about TLS and structure from motion address the question: "better describe why users would want to use these systems versus techniques like photogrammetry"

Structure from motion (SfM) especially from Unmanned Aerial Vehicle (UAV), is another tool to derive 3-dimensional (3D) point clouds for geoscientific applications. For example, Lague (2020) compared SfM to TLS for fluvial geomorphology and Wilkinson et al. (2016) for digital outcrop acquisition. Its main advantages are low-cost, uniform point density, high resolution RGB information with the limitations of no penetration through vegetation, requirement of good Global Navigation Satellite System (GNSS) signal, and UAV flight regulations. In addition, SfM is problematic for surfaces with homogenous textures like snow and is limited to well-lit environments. In caves SfM has been used with digital close-range photogrammetry by a digital single-lens reflex camera mounted on a tripod in the study of Pukanská et al. (2020), where they also compared it to TLS data. Their conclusion was that both methods have their specific requirements, advantages, and disadvantages. Therefore, they recommended a combination of both methods for mapping complex cave spaces. „

For details please also see the attached latex diff which shows all the changes.

Once again, many thanks for the valuable input and all the best – on behalf of the author team,

Thomas Gölles

Specific Comments:

Line 19: Explain why you used such a long-range lidar for such confined areas?

We used the long-range TLS as a reference since it is a state-of-the-art TLS with high accuracy and high precision. The VZ6000 has a minimum rang of 5 meter and is therefore suitable for static compression in section 3.2.

We added "high accuracy and high precision" to the text for further reasoning.

24: This is a minor comment, but I don't see how you get the "LI" in MOLISENS. Don't you need to put "Lidar" in there, for example, Mobile Lidar Sensor system.

This was a typo. The sentence now includes lidar.

28: Here and elsewhere, when you say automotive lidar, it is a little unclear. What you mean is that you are using small lidar sensors that were originally designed for the automotive industry. But it is a little confusing because people might think you mean lidar mounted on an automobile. So try using a more descriptive term like lidar developed for the automotive industry.

This is a very valid point and we added it specifically in the title:

This is now also reflected in the improved title: "MOLISENS: A MObile LIdar SENsor System to exploit the potential of small industrial lidar devices for geoscientific applications "

The vehicle independence is also mentioned in the first paragraph of the introduction and the abstract. Therefore, it should now be clear that it is a lidar sensor from the automotive industry which does not require a vehicle in our setup. Therefore, we kept the terms automotive lidar throughout the text to distinguish between to Ouster lidar and "conventional" TLS such as the Riegl VZ6000.

43-44: I think you should avoid things like prices and saying when an instrument is expected to be out. Here you write "late 2022", but what if your paper isn't out in 2022? What if the instrument isn't ever released.

Yes that's true, it is speculative and hence  we deleted the sentence.

145: Can you explain how the GPS works in a cave?

It does not. The system does not depend on continuous GNSS data, as mentioned in line 242 in the original version of the manuscript.

Since the section deals with the software stack and MOLISENS is for multiple applications, and not just caves, we think it is appropriate to leave this information further down the line in the SLAM algorithm section.

206: The sentences here: "We found that…" should be in the results.

We moved the sentence to section 6.1

218: Where you have written ( etc) what are the other things that you are measuring? I'm more interested in that, than the information in Table 1. Consider putting those measurements in a table.

We now list all attributes which are stored by the Ouster sensor.

„In addition to the x-,y-, and z- coordinates, also range, intensity, reflectivity, ambient near infrared, azimuth angle, and time stamp for each point are stored"

226: where you say "How many clusters of points in a 0.5m radius exist between 5 and 10 in THE x direction …" What are the units you are refering to when you say between 5 and 10?

Added: „between 5 and 10 m"

345-361 Add this information to the intro.

Done.

Figure 1. Add labels (a, b, and c) for the sub-figures here. Also in (b) consider adding something for scale, such as a pen.

Done. In addition, the caption was adapted and the presentation of the battery was made consistent with the rest of the image.

Figure 5. Add a colorbar in 5d.

Done.

Figure 6. Use a more complete sentence and be more descriptive in the caption for (a). (b) Add a colorbar, label the hand-rail, and try using something like the EDL filter to make the point cloud more visible in the figure.

We added the color bar, hand-rail label, and increase the readability of the point cloud image.

The new caption reads as follows: „Panel (a) shows karst shapes, concrete, and metal structures in the Lurgrotte cave (image by Christian Bauer). Panel (b)  shows a detailed image of the cave section including the walking path and handrail (created with CloudCompare)"

Figure 7: Show location map of where the glacier is located (similar to inset in 5a). Add a colorbar to all figures.

We added a detailed map and completely reformatted the figure. All point clouds now use the same color bar.

Table 1: Is this table necessary? The specs feel somewhat out-of-step with a journal article.

The second reviewer was interested in the specs, and we think that it is of importance for users, since they enable many new applications which are otherwise impossible with a heavy and bulky instrument. Therefore, we deleted Table 1 but mentioned the key specs in the caption of Figure 1.

---

## Author Comment (AC2)

**Referee 2**

**General Comments:**

This is an interesting paper presenting a methodology for developing an automotive lidar unit for use as a mobile lidar unit with geoscience applications. The manuscript does a nice job of explaining the components of the system and the testing that was done to find the applicable range and resolution. Perhaps the one component that I feel the paper is missing is a comparison of a point cloud results from the two test areas with a point cloud generated from the Riegl scanner used for the initial testing. The authors do highlight why this would be impractical for the entire area at the Lurgrotte Semriach, but even a small section would provide useful evidence for how comparable the mobile unit is to a more well-known tripod-based unit in a complex natural setting.

Dear Editor, dear reviewers,

We are very grateful for two very detailed and constructive reviews and appreciate the valuable time put into this. We believe by incorporating the reviews we managed to achieve a much more mature manuscript which we hereby submit for your consideration.

In the following we mark black the comments given by the reviewers and the editor, give our answers and comments in blue and indicate how we addressed the amendments in the manuscript in green.

Unfortunately, we had technical difficulties with MOLISENS during the measurements at Lurgrotte Semriach which were later solved (data was written on SD card not SSD and overheating of the data logger). Therefore, we have no MOLISENS data in the area where we have VZ6000 data. Nevertheless, we have a formal evaluation in section 3.2 which gives insight in the differences between the devices. We also wrote about drifts for cave measurements which still needs to be quantified already in the original version of the manuscript.

„The lack of GNSS data for cave measurements caused drifts induced by small propagating errors in the IMU data. These drifts are yet to be quantified."

For details please also see the attached latex diff which shows all the changes.

Once again, many thanks for the valuable input and all the best – on behalf of the author team,

Thomas Gölles

**Specific Comments:**

Line 24: remove "allows to build" and replace with "builds"

Done

Line 33: Three systems are mentioned (Lidar, Radar, and Camera) so remove the word "both"

Done

Line 87: I may be misunderstanding something, but here you note that the system can be powered by AC/DC adapter, but in line 91 you note that it requires external batteries to be mobile. If the appeal of this system is that it is mobile doesn't that exclude the AC/DC power option.

The system works with both: battery and power adapter. For testing and long-term observations, the power adapter option is very useful. Therefore, we left this part unchanged in the new version of the manuscript.

Table 1: I know the previous reviewer mentioned removing this table, and while I agree it may not be necessary, I do appreciate knowing the size and weight of the unit, it is much lighter than many others available, and this certainly increases the number of potential users.

We deleted Table 1 but mentioned the key specifications in the caption of Figure 1.

Line 94: please consider writing out your acronyms, in particular IMU. It was easy enough to find what this was, but for those of us not familiar with these acronyms, writing them out can be helpful for clarifying their purpose.

We use the LaTeX package glossaries, which automatically writes the long version when an acronym is used. We deleted some single use and not commonly used acronyms to reduce the number of acronyms. Also, we now use GNSS instead of GPS consistently.

Line 100: It appears that HAT is typically a capitalized acronym. Also, this could again be written out once to help clarify what it is.

This acronym was missing has been added to the manuscript.

Line 217: here it is noted that the sensor records point cloud information, yet earlier in line 173 you noted that the lidar doesn't produce the point cloud, but rather the raw data are timestamp, measurement ID, and range. I suspect these are discussing two separate steps, but there is some additional information may help clarify, as both lines appear to be discussing the Lidar sensor.

The lidar OS1-64 could also produce point cloud information directly. We changed the setup to only record raw data. We record raw data with the data logger to save storage space. Therefore, this needs a separate step to get the point cloud data with data associated to each point.

We added the following clarification into the text that we do this to store more data:

„We decided to record only raw lidar data to be able to store more data at about 1.3 million points per second…".

Line 289: I am curious about the errors in ice. Perhaps these errors are well covered in the literature and there can be citations, but if not, maybe there can be a few more

details. I would imagine that because light can move through ice it might create some errors in the data. Perhaps this is accounted for in the lower return intensity described later, but it might be helpful to address outright how the MOLISENS system compares to others when surveying ice.

These errors are not well documented in the literature. The Ouster unit has a wavelength of 855nm as compared to the 1064nm of the VZ6000. Therefore, the reflectance of ice is higher for the Ouster unit which should potentially improve the performance. As can be seen in the following figure from my PhD thesis (https://www.researchgate.net/publication/321526677_Impurities_of_glacier_ice_accumulation_transport_and_albedo).

[Figure]

**Figure 1.3:** **(a)** spectral irradiance at the top of atmosphere (TOA) and sea level (ASTM International, 2012). **(b)** real part and **(c)** imaginary part of the complex refraction index. The extinction coefficient ($k_\lambda$) of ice is strongly dependent on wavelength, and is very low in the visible spectrum (Warren and Brandt, 2008). The extinction coefficient of black carbon (BC) does not depend on wavelength, and the recommended value is 0.79 (Bond and Bergstrom, 2006). The exact value of BC depends on the source and range of the values is indicated in grey. Dust has a lower $k$ than BC, shown here in examples from the Sahara (Burkina Faso, Wagner et al. (2012)) and Asia (Zhangye: 39.082°N, 100.276°E, Ge et al. (2010)).

The sentences read as follows:

"Some outlier points are visible that might be a result of the scanners range errors and the torso of the person holding the scanner. In general, the Ouster OS1-64 should perform well for glaciological applications due to the wavelength of 855nm where absorption in ice is

lower (Warren and Brandt, 2008) than at the wavelength of the VZ-6000 at 1064nm or other TLS which have a wavelength of typically 1550nm (Deems et al., 2013). Further investigations of the errors are needed in the future."

Deems, J. S., Painter, T. H., and Finnegan, D. C.: Lidar measurement of snow depth: a review, Journal of Glaciology, 59, 467–479, https://doi.org/10.3189/2013JoG12J154, 2013.

Warren, S. G. and Brandt, R. E.: Optical constants of ice from the ultraviolet to the microwave: A revised compilation, Journal of Geophysical Research: Atmospheres, 113, https://doi.org/https://doi.org/10.1029/2007JD009744, 2008.

Line 294: There appears to be a few words missing after "These are the price, size, weight, and robustness...", or perhaps there should be a semicolon connecting this with the previous sentence.

The sentence now reads: "These advantages are the lower price, smaller size, lower weight, and increased robustness but also the ability to acquire data in narrow spaces"

Line 324: "A useful" not "an useful"

Done

Line 329: These sentences seem to imply that the lower intensity returns for ice surfaces may be better for change detection. If this is true, could you provide more information as to why. If this isn't meant to be implied, consider removing the word "hence", and possibly moving this line.

We added a new sentence which could clarify what we meant by that:

„In general, surface types could be distinguished if they have a significant difference in intensity."

Line 335: Does this line suggest that Structure from Motion (SfM) would be a preferred method of monitoring coastal bluffs? This is the first mention of SfM, it might be helpful to have a line or two about the advantages of this system in comparison that one, particularly where you might have good GPS control.

We added a completely new paragraph in the introduction mentioning SfM and how it compares to TLS. Here, we also use the acronym 'SfM' since it has been introduced in the introduction.

Line 336-339: This sounds amazing, I look forward to this technology becoming more widely accessible.

Yes.

As a general comment about the discussion and conclusions, river systems are mentioned a few times as an application for this technology, but I suspect the laser isn't powerful enough to penetrate through water. Similarly, there is no mention of multiple returns, so I suspect this isn't penetrating through vegetation. It may be worth noting these caveats specifically depending on the anticipated audience.

It is true, that the laser is not powerful enough to penetrate through water but indeed, it can be used to map or monitor the dry parts of a riverbed. We added the following sentence to section 6.1:

„Another challenging surface is water, which absorbs typical wavelengths of automotive lidar in the near- infrared (e.g. Lague, 2020). „

We added two sentences about the number of returns of automotive lidar in the introduction: "Most state-of-the-art automotive lidars provide a single return or dual returns, while some have up to five returns (Ocular Robotics Limited, 2018) and one which offers full-waveform information (LeddarTech Inc., 2021). Therefore, this limits the application of automotive lidar where multiple returns are needed, as for example for vegetation removal. „

---

## Author Response (AR2)

**Referee 1**

**General Comment**

In general, I think the authors have done a good job revising this manuscript and I do not have any further major recommendations. I do have a few small suggestions.

Dear Referee 1,

We are very grateful for your constructive reviews and appreciate the valuable time put into this. We believe by incorporating the reviews we managed to achieve a much more mature manuscript which we hereby submit for your consideration.

In the following we mark black the comments given by the reviewers and the editor, give our answers and comments in blue and indicate how we addressed the amendments in the manuscript in green.

**For details please also see the attached latex diff which shows all the changes.**

Once again, many thanks for the valuable input and all the best – on behalf of the author team,

Stefan Muckenhuber

**Specific Comments**

-first response uses the term 'Non-automotive' again, I think you should consider taking the term automotive fully out of the document. I like your 'small industrial' description better than automotive. Consider this example. If I took a camera, that was originally designed for a phone, and mounted it on a circuit board it would be strange to call it a phone camera. It is no longer related to the phone at all, the only relationship is that the part was originally designed for use in the phone industry. I think it's a similar situation with your lidar. Its not on/in a car and so it really isn't related to automobiles at all.

We agree and have replaced the term 'automotive' at several places with the term 'small industrial'. We have left the term 'automotive' only at places where it is relevant to refer to the automotive industry, e.g., as driver for new lidar technology.

-If you want to show price comparisons, then I think you need to show both the price of the new proposed unit and the price of the Riegl units. Price is something that tends to make little sense after about a decade. Consider the drop in the British pound before/after Brexit, or price inflation from 2019-2022.

We agree and added a price estimate for the TLS. NB: the Riegl VZ-6000 costs currently around 160,000 EUR.

It should be cheaper than TLS (currently around 100,000 EUR to 200,000 EUR), with a high sampling frequency and be independent of natural or artificial light, while being small and light enough to carry into tight spaces like caves.

-As the other reviewer pointed out, the authors only really compare the Riegl and the MOLISENS in a lab environment, they don't compare it in the field. I think that's fine, but I think they need to talk about it in that way. For example, take line 18:
"To evaluate the performance of MOLISENS, we present a comparison between the integrated automotive lidar Ouster OS1-64 and the state of the art, high accuracy and high precision TLS RIEGL VZ-6000. "
And change to:
"To evaluate the performance of MOLISENS, we present a comparison between the integrated automotive lidar Ouster OS1-64 and the state of the art, high accuracy and high precision TLS RIEGL VZ-6000 in a set of controlled experimental setups. We then use the Ouster OS1-64 in several real world settings …. "

We agree and changed the corresponding sentences in the abstract to the following:

To evaluate the performance of MOLISENS, we present a comparison between the integrated small industrial lidar Ouster OS1-64 and the state of the art, high accuracy and high precision TLS RIEGL VZ-6000 in a set of controlled experimental setups. We then apply the small industrial lidar Ouster OS1-64 in several real world settings.

**Editor**

**Comments:**

Dear Dr. Goelles and co-authors,

Thank you for your handling of the paper and response to referee comments. Reviewer #1 has responded positively to this and suggested that the paper be published as-is, but has also added some of their own comments. Referee #2 has contacted me to note that they cannot commit to a full re-review, but leave the comment that they are not sure that your edit about return intensity answered their question. In particular, Referee #2 wrote to me that,
"I was more interested in the suggestion that a lower intensity return would be better for change detection. I find that counterintuitive. That being said it may be an interpretation issue because of how the sentence was written."
If you can address these minor comments, I would be happy to take a look at your handling of them myself, sparing us another round of review.

Best wishes,
Andy Wickert

Dear Editor,

We are very grateful for your constructive reviews and appreciate the valuable time put into this. We believe by incorporating the reviews we managed to achieve a much more mature manuscript which we hereby submit for your consideration.

In the following we mark black the comments given by the reviewers and the editor, give our answers and comments in blue and indicate how we addressed the amendments in the manuscript in green.

**For details please also see the attached latex diff which shows all the changes.**

Once again, many thanks for the valuable input and all the best – on behalf of the author team,

Stefan Muckenhuber

**Specific Comments**

Referee #2 wrote to me that, "I was more interested in the suggestion that a lower intensity return would be better for change detection. I find that counterintuitive. That being said it may be an interpretation issue because of how the sentence was written."

We agree that there can be an interpretation issue because of how the sentence was written. Lower intensity values itself are not better for change detection. Our intention was to point out, that the intensity difference between ice (low intensity values) and glacier surfaces covered with, e.g., moraine material or sediments (higher intensity values) allows to distinguish between bare ice and surface covers such as moraine material or sediments. We changed the corresponding sentences in the manuscript to the following:

In addition, different surface types can be distinguished if they show a significant difference in intensity. E.g., the recorded intensity values for ice surfaces are significantly lower than for surfaces covered with, e.g., moraine material or sediments. This allows to distinguish bare ice from gravel based surface covers.